# Health-aware Point of Interest Recommender system

Ivan Kropotov[+]
[+]University of Helsinki

**Abstract**

In modern cities, a wide array of recreational facilities, such as outdoor gyms, skating rinks, and sports halls, are available, with the city aiming for their maximum utilization. Concurrently, the government's objective is to promote healthy lifestyles and enhance citizens' overall wellbeing. To address both objectives, we present a prototype recommender system designed to recommend facilities that promote well-being and consider users with various medical conditions, e.g., back pain or mental stress. These recommendations are influenced by medical guidelines, current weather conditions, and the user's location. Our user-friendly interface facilitates preference specification and visualizes recommendation results. Additionally, the system also suggests optimized routes that consider air quality in the city.

## 1 Introduction

The city of Helsinki provides public amenities that provide physical activities to the residents and visitors of the city. Importantly, the city wants to keep the utilization rates of its investments as high as possible, but finding interesting places is often difficult due to the number of options to choose from. Another interest of the city is to keep its residents healthy and to improve their quality of life, which also benefits the residents. One way to achieve this is to encourage people to use the provided facilities for physical activity. The city of Helsinki has a service for searching for the amenities called Service Map [1], but it lacks the recommendation functionality making it less suitable for discovery of new venues.

To address the issues, we developed a prototype recommender system which suggests points of interest (POI) where a person can partake in physical activities, e.g., gyms, sport centers, etc. One of the main properties of the system is that it is designed to provide sound recommendations to people with various medical conditions, like mental stress or asthma, based on medical recommendations. Other aspects that affect the recommendation results include the current weather, location of the user, point of interest information, and the user's activity preferences. In addition to the POI recommendation, our system is capable of recommending walking routes to the amenities and round trips, that have the best air quality along the route, akin to Green Paths [2].

The supporting evidence for our prototype is quite limited. We have not conducted rigorous user experiments, but we have shown the system to multiple people, including city of Helsinki officials, whose feedback was in general positive.

## 2 Goals

There two main goals of the project were to make people more aware of the physical activity facilities the city of Helsinki provides and to improve the user's well-being by suggesting healthy activities. Both goals could be achieved by designing a suitable recommender system, which we attempted in this project.

The objective of raising the utilization factor of the city's physical activity amenities is a very important one from both the city's and the resident's point of view. On the one hand, the city doesn't want to see its investments into the infrastructure go to waste, especially due to lack of awareness. Additionally, it is very important that people use the services that the city has built, so that the city officials will get enough feedback on the usefulness of the services. This directly benefits the residents, since in the future the limited budget can be directed to the services and infrastructure that people need and want to use.

Improving the well-being of the residents also benefits both the people as well as the city/government. At the level of a single person, it is obvious that adopting healthy habits and hobbies helps the person to live a better life. The city, on the other hand, benefits from the healthier population in the way of reduced medical and social costs. Thus, investing a comparatively small amount of budget into facilities for a person to use can save from paying costly medical fees.

## 3    System Architecture and Design

### 3.1    Hardware

Our prototype is purely software based.

### 3.2    Software

Since our system was easily divided into multiple well-defined components, we went with a modular, microservice - like, architecture. In total there are four components, each having its own function and residing in their own Docker containers. This architecture made it easier to develop and test different modules due to them being in separate containers that could be rebuilt individually when needed. For the simplified architecture overview see Figure 1.

The main, or at least the most visible, component is the user application. This component provides the UI through which the user interacts with the system by setting the preferences and exploring the recommendations. It also handles communication between the other components and has some logic related to recommendations. The component itself is built using Python, with the UI being developed using Dash framework [3], since it provided powerful but simple to use visualization components.

The main data storage, i.e., the POI database, is handled using MongoDB. This database is populated with data from the Service Map API [4] which is scraped and preprocessed to help with the recommendation step. The preprocessing and scraping are done using a Python script, with two main preprocessing steps being performed. The first one was adding tags to the POIs that correspond to types of activities, e.g., aerobic exercise, strength exercise, outdoor activity, etc., that were primarily used for finding suitable activities for different medical conditions and weather-based recommendations. The other important preprocessing step was building a geospatial index in the database to enable efficient location-based recommendations.

The third component is the routing engine which provides navigation and route suggestions. For this project we use an open-source routing engine called Graphhopper [5], due to its speed, extendibility, and good documentation. To incorporate air quality index data (AQI) into the route suggestions, we had to modify the code slightly, so our system uses a forked version of Graphhopper. The way that the routes with good air quality are found is using environmental impedance function, which is the same as in Green Paths [2]. The routing engine allows for building routes for to the points of interest as well as suggesting round trips for a refreshing walk.

The last component of the system is a collection of scripts running in a separate Docker container and handling background tasks. These tasks include setting up MongoDB and Graphhopper routing engine when the app is launched, downloading and AQI data from the Finnish Meteorological Institute (FMI) and keeping it up to date, etc. The scripts in this component are mainly Python and shell scripts.

## 4    Addressing Challenges

There were a few challenges related to this project, most of which were small, but there were a few standouts. The first one was the availability of the medical recommendations of physical activities for various medical conditions. Initially we thought that good guidelines, which we could use as a baseline of a prototype, could be found in medical literature. We did manage to find some highly regarded medical papers on the topic, but unfortunately, they were either too general or too specific for our use case. In the end, we

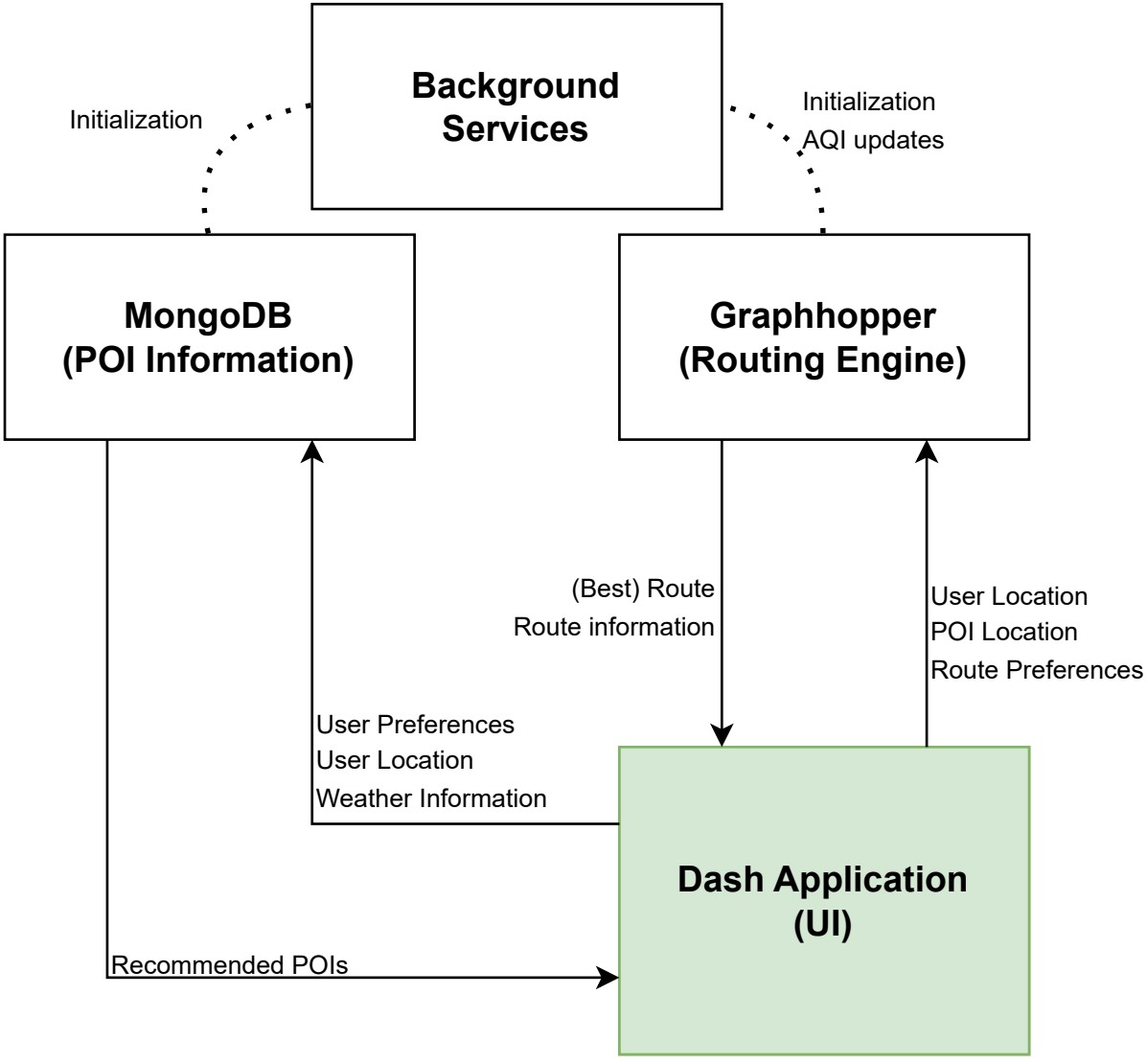

Figure 1: System architecture diagram.

gathered as much of the information as possible, and created a set of activity categories that correspond to those in the papers. This served well as a baseline, but in the future, we plan to use medical professional's help in finetuning the recommendation process.

The next big challenge was related to the usage of air quality data for routing. Initially, we were planning on using the Green Paths [2] public API for routing based on air quality. However, we found that the service was a bit too slow for our purposes and it was complicated to extend for some of the new capabilities that we were planning on having in our system, e.g., round trips. We solved this issue by using an open-source routing engine called Graphhopper [5], which was fast and well documented, and had built-in capabilities for, among others, round trip generation and wheelchair profiles. Unfortunately, adding air quality to the criteria with which the routes are searched for wasn't supported in the available configuration and we had to modify the software library a bit. In the end, we forked the Graphhopper repository and extended its functionality by implementing in Java a similar algorithm to that in Green Paths.

## 5   Performance Evaluation and Testing Results

Properly testing our system would require a user study, which unfortunately was not conducted due to the lack of time and funding. The prototype system has been shown to our contacts in the city of Helsinki who gave us positive feedback, however this information is hardly sufficient as a rigorous proof of the perfor-

mance of our system.

## 6  Concluding Remarks and Avenues for Future Work

All in all, this was a fairly challenging project, during which we learned a lot. We had to learn some geoinformatics and brush up our Java coding skills, but we managed to overcome most of the challenges. We regret that we didn't have a chance to conduct a real user study, which will be left for the future work.

In the future, we plan on continuing developing this prototype to make it into a more useful product. First, we intend to improve the recommendation quality of the system by working with healthcare professionals on creating better guidelines for different groups of people, e.g., young, old, healthy, etc. We also want to incorporate more information about the physical environment, with plans on adding more accurate weather and air quality information. Data on road surface, inclination, and other data helpful with people with reduced mobility will also be incorporated in the future. We also plan on adding the geolocation of the user and providing detailed walking instructions. And of course, we want to keep increasing the number of points of interest and make them more varied than now. Improving the user interface is also high on our list. The aforementioned user study is an integral part of our future work as well.

## 7  Availability

The video demonstrating our system can be found at `https://helsinkifi-my.sharepoint.com/:v:/g/personal/ivkropot_ad_helsinki_fi/EaLhEU9VUztDrx3PJHSw0jAB5c3IIejDdj4iV8ijWTk1Sw?nav=eyJyZWZlcnJh e=pN9OQx`.

Our recommender system can be accessed here `http://86.50.230.152:5000/`

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
