# OpenReview forum: "Health-aware Point of Interest Recommender system"
_helsinki.fi/ESPC/2023/Competition — ESPC 2023 ShortPresentation_

### Official Review · Reviewer_AzF4 · 2023-11-14

**Rating:** 3
**Confidence:** 3

**Summary:**

Article presents the recommendation system providing points of interest information based on the user health condition. The system incorporates also real-time weather and air quality data.

**Strengths:**

1. The system is fully designed and implemented.
2. System uses both static data (like Points of interest, location information) and real-time data (like weather and air quality).
3. System demonstration is provided with the video recording.
4. Author provides limitations of the work and highlights future directions.

**Weaknesses:**

The work is nice and could be discussed during the event. One comment to the report:
1. Would be beneficial if the motivation could be accompanied with the related work. Why this solution is needed? How is it different from the related ones?

---

### Official Review · Reviewer_wDfE · 2023-11-17

**Rating:** 1
**Confidence:** 3

**Summary:**

In this report, author has presented a software based system for providing recommendations for visiting certain places along with optimized routes considering the air quality.
There are no sensors used.
Existing APIs are used for maps and air quality index.

**Strengths:**

A software system prototype is developed with an aim to help people stay healthy and reach places that help promote healthy lifestyle.
I think this is just a usecase of this prototype and it can be extended to any other application as well.

**Weaknesses:**

The idea is very simple and trivial.
More details are needed to appreciate the challenging aspects of this work
No actual sensing is done.
No evaluation is done.
The project is in naive stages and needs a lot of work to be presented.

---

### Official Review · Reviewer_SBqv · 2023-11-20

**Rating:** 2
**Confidence:** 4

**Summary:**

A purely-software based recommender system is designed to make people more aware of the physical activity facilities the city of Helsinki provides and to improve the user’s well-being by suggesting healthy activities. The tools used are python, MongodB, serviceMap APIs and graphhopper along with dashboard application and appropriate background services.

**Strengths:**

The project takes into account medical condition, air quality and city infrastructure to suggest different activities. This is very useful.

**Weaknesses:**

1. The system is not tested and evaluated. This is the biggest drawback.
2. It is also not clear why this approach is the best.
3. Novelty in the system is not clear.
4. As pointed by the authors, the medical recommendation for different medical conditions may not be easy as they differ from case to case.